# The Effect of COVID-19 Lockdown on PHQ Depression Screening Scores for High School Athletes

**DOI:** 10.3390/ijerph19169943

**Published:** 2022-08-12

**Authors:** Devin P. Adams, Jasmine R. Holt, Jenna A. Martin, Danielle M. Houpy, Kathryn A. Hollenbach

**Affiliations:** 1Transforming Mental Health, Rady Children’s Hospital, San Diego, CA 92123, USA; 2Medical Practice Foundation, Rady Children’s Hospital, San Diego, CA 92123, USA; 3Department of Pediatrics, University of California, San Diego, CA 92093, USA

**Keywords:** COVID-19, high school athletes, depression screening, PHQ, lockdown, pandemic

## Abstract

Adolescent behavioral health was in crisis before COVID-19. The shutdown and reopening of in-person learning and extracurricular activities may have worsened this crisis. We examined high school athletes’ depression before and during the pandemic. Data were collected as part of a pilot program incorporating Patient Health Questionnaire (PHQ) screenings during high school sports physicals before the COVID-19 lockdown and three timepoints after. Statistical comparisons were made using logistic regression. A total of 927 individual scores were analyzed: 385 from spring 2020; 145 from fall 2020; 163 from fall 2021; and 234 from spring 2022. Fall 2020 students were 3.7 times more likely to have elevated PHQ-2 scores than spring 2020 students (95% CI = 1.8, 7.6). Fall 2021 and spring 2022 scores did not differ significantly from pre-pandemic, although trends of elevated scores persisted (OR = 1.6; 95% CI = 0.7, 3.5, and OR = 1.2; 95% CI = 0.6, 2.4, respectively). A significant difference in PHQ-9 depression severity classification was detected over time (*p* < 0.01). Elevated PHQ scores were found after the onset of the COVID-19 pandemic. After the initial peak in fall 2020, scores decreased but did not reach pre-pandemic levels.

## 1. Introduction

Countermeasures to the spread of COVID-19 led to major changes in daily living to mitigate the global loss of life [1]. While stay-at-home orders and social distancing were implemented to decrease virus spread and protect physical health, the effect these mandates had, and continue to have, on mental health is still being investigated. Globally, there have been reports of declining mental health among the general population due to lockdowns [2,3,4,5,6,7,8]. This is consistent with previous research documenting adverse mental health outcomes following outbreaks or natural disasters [9,10,11,12,13,14]. Even before the pandemic, a trend of increasing demand for the treatment of internalizing disorders among adolescents had been observed in the United States (US) [15]. Initial research since the onset of the pandemic has documented adverse mental health effects associated with the lockdown and social distancing mandates among young adults and adolescents in the US [16,17,18,19]. It is important to fully understand this observed mental health decline, and particularly how far the decline in adolescents has progressed, as well as if this decline is beginning to plateau or even reverse.

One subgroup of adolescents that warrant further investigation is high school sport participants. School closures resulted in the cancellation of all, or most, extracurricular activities and these students may have lost more than simply the social interaction from school classmates. Given that physical exercise has shown to be protective against depressive symptoms in adolescents [20], the loss of organized physical activity may have amplified the effects of the shutdown [21,22]. Limited studies have focused on student athletes, but research has shown the benefits of increased physical activity level, social interactions, and sports participation on overall mental health functioning [23,24,25,26].

Data from a pilot program that incorporated depression screening in high school athletic physical exams at a large suburban high school was used. Utilizing this previously collected data allowed us to compare athlete depression screening results before and after the COVID-19 pandemic lockdown was implemented. These screenings are part of the school sponsored physicals that student athletes voluntarily attended to obtain medical clearance for high school sports participation.

## 2. Materials and Methods

### 2.1. Depression Screeners

The Patient Health Questionnaire-2 (PHQ-2) and Patient Health Questionnaire-9 (PHQ-9) are screening tools for depression that can be self-administered and are validated for use in adolescents [27,28,29,30]. The PHQ-2 was developed as a brief depression screening tool to accommodate high patient volume clinical settings [27]. It contains the first two questions of the PHQ-9, each scored on a scale of 0 to 3 for a maximum score of 6 [27]. PHQ-2 scores ≥ 3 have a sensitivity of 83% and a specificity of 92% for major depression when compared to outcomes of independent mental health professional structured interviews [27]. The PHQ-2 has also shown to have high internal consistency [31,32].

The PHQ-9 is also self-administered, consisting of nine screening items developed to assist in making a diagnosis of a depressive disorder [30]. Scores on the PHQ-9 range from 0 to 27 with indication thresholds at scores of 5, 10, 15, and 20 to represent mild, moderate, moderately severe, and severe depression, respectively [30]. As an instrument to detect depression in a primary care setting, the PHQ-9 has an overall sensitivity of 88% and specificity of 88% for scores ≥ 10 [30]. The PHQ-9 has also demonstrated high internal consistency [30,31,32]. In the sample for this study, the Cronbach’s alpha for the PHQ-9 was 0.83 indicating good reliability. Both the PHQ-2 and PHQ-9 have been used effectively to monitor outcomes over time [27,30].

### 2.2. Data Source

This study utilized deidentified previously collected data. The analysis of these deidentified data was approved by our institutional review board. The school health program through which the data were collected obtained consent for school sports physicals, which included the depression screening. The students in this data include adolescents between the ages of 13 and 17 who intend to obtain medical clearance for participation in high school sports. As part of the school physical, students were given the PHQ-2 on paper to fill out. After completing the PHQ-2, a monitor collected and immediately scored the screeners. Students scoring less than 3 were excused and students scoring 3 or higher were identified as positive on the PHQ-2 and given the PHQ-9. PHQ-9 scores were scored immediately, and students were directed to their next station based on score: scores of 0 on question 9 (assessing suicide risk) in conjunction with a total score between 0 and 9 were given information about mental health and depression. Scores of 1 or higher on question 9 or a total score higher than 10 were seen by a licensed clinical social worker for further assessment of suicide and depression risk, as well as developing an appropriate treatment or referral plan.

#### Screening Timing

The initial screenings in spring 2020 occurred one month before the state’s COVID-19 countermeasures went into effect. The second screening in fall 2020 occurred when a delayed start to organized high school athletics was planned in the academic calendar. There were also significant COVID-19 safety protocols implemented based on level of contact within each sport. The third and fourth screening, fall 2021 and spring 2022, respectively, occurred when high school sports had resumed playing its normal seasonal schedule.

### 2.3. Analysis

Deidentified PHQ-2 and PHQ-9 scores along with limited student demographics were entered into a database and made available for analysis. To improve accuracy of self-reported demographic information, the demographics portion of the survey was simplified from free response questions to categorical check-box responses with the option to write in if categorical options were not appropriate. This led to a shift in demographics between fall 2020 and fall 2021. To address repeated measure concerns, randomly selected scores for students who attended multiple sessions were used for analysis. Data were analyzed using STATA 16 (College Station, TX, USA) [33]. Initial analyses used fisher’s exact tests to compare PHQ-2 and PHQ-9 results at different time points: pre-pandemic (spring 2020) and post-pandemic (fall 2020, fall 2021, and spring 2022). Initial bivariate analyses were conducted to assess demographic differences and PHQ results by timing of assessment using chi-square and Fisher’s exact tests. Logistic regression was used to determine the odds of elevated PHQ scores post pandemic lockdown using the pre-pandemic data as reference. Changes to PHQ-2 scores were assessed by evaluating the change in the number of students scoring at least 3 or higher across the four time periods. Potential confounding factors included age, grade, team vs. individual sports, and timing of assessments. Adjusted odds ratios and 95% confidence intervals (CIs) were calculated. To assess the difference between pre-pandemic (reference) scores and scores from each of the post-lockdown assessments, timing of assessment was modeled comparing each post-pandemic assessment to the pre-pandemic assessment. Variables retained in the final model were independently associated with elevated PHQ scores.

## 3. Results

After accounting for repeated measures, data from 927 students were analyzed. For the PHQ-2 scores, there were 385 in spring 2020, 145 in fall 2020, 163 in fall 2021, and 234 in spring 2022. Demographic characteristics of students sampled at each period are presented in Table 1.

### 3.1. PHQ-2

The proportion of those having elevated PHQ-2 scores was lowest in Spring 2020, peaked in Fall 2020, and was still higher in Fall 2021 and Spring 2022 than it was pre-pandemic (Table 2). A test for trend identified a significant change in PHQ-2 score severity classification across time periods (*p* = 0.03). The proportion of students scoring 3 or higher on the PHQ-2 was 6% in the pre-pandemic group, then rose to 14% in fall 2020, and has been at 9% for fall 2021 and spring 2022. Students who attended in fall 2020 had a 3.7 times increased odds of scoring 3 or higher on the PHQ-2 (95% CI = 1.8, 7.6) compared to students in spring 2020 (pre-pandemic) (Table 3).

### 3.2. PHQ-9

There were PHQ-9 results available for analysis from 25 students assessed in spring 2020, 18 in fall 2020, 14 in fall 2021, and 23 in spring 2022 (Table 2). The median pre-pandemic PHQ-9 score was 7 (5, 15) (25th, 75th percentile) in spring 2020 and increased with each successive testing period. The proportion of students scoring 10 or higher was 36% in spring 2020, 44% in fall 2020, 57% in spring 2021, and 60%, in fall 2021. A significant difference in PHQ-9 depression severity classification was detected (fisher’s exact (*p* < 0.01). The change in distribution of PHQ-9 scores after the onset of the pandemic identified a greater proportion of scores persisting at the moderate and the moderately severe depression severity classification.

## 4. Discussion

The arrival of COVID-19 in the United States produced far-reaching uncertainty as well as profoundly impacting communities nationwide. With sports programs playing a large role in many student athletes’ lives, the suspension of these programs created additional hardships for at least some of these adolescents during an already stressful period. Our findings identified a significant increased odds of depression demonstrated by elevated PHQ scores among high school athletes associated with timing of the COVID-19 pandemic lockdown. Although not statistically significant, the elevated odds ratios for the fall 2021 and spring 2022 PHQ-2 scores indicate that this trend may be persisting over time. While these changes coincide with the timing of the COVID-19 lockdown, other factors beyond school athletics could have influenced the changes in screening scores.

We also found that female gender and Hispanic ethnicity were independently associated with increased depression risk. Sub-analysis of data comparing only the initial lock-down period (fall 2020) to pre-pandemic spring 2020 did not find gender or ethnicity to be independently associated with elevated PHQ-2 scores. Only the timing of assessment was found to be associated with elevated scores. This may reflect the overwhelming effect timing of the initial lock-down had on all student athletes. This adverse effect could also have been further fueled by the lack of understanding and general anxiety surrounding COVID-19 during the initial lockdown. It is encouraging that although the rates of depression remained elevated relative to the base-line pre-pandemic rate, the rate did decrease from the initial post-pandemic measure and may be indicative of resiliency among high-school athletes associated with or independent of their return to sports. Future studies of high school athletes, and adolescents in general, are warranted to understand how they have and continue to respond to the everchanging pandemic.

For the overall PHQ-9 scores, a consistently increasing median over time is concerning but our limited sample size prevented further analysis. The PHQ-9 score median changed depression severity classification from mild to moderate depression. Although a statistically significant shift in PHQ-9 score distribution was noted, no student scored 20 or higher (severe depression classification) in the three post-pandemic screenings.

Consistent with trends noted in the recent literature, the timeline of the lockdown has been associated with declining mental health of this subgroup of adolescents [21,22,23,24,25,26]. Given the temporal sequencing of the PHQ depression screening results among the high school students and the mandated lockdown, it is logical that the increased rate of depression observed is associated with the pandemic.

Our findings are limited to a single high school and, therefore, may not be generalizable to other schools. Some students may have received their school athletic physical from their own physician, so these results may not be representative of all student athletes at this or other schools. The voluntary nature of the screening event, as well as other potential outside factors such as fear regarding large crowds, may have discouraged students from attending the event post initial COVID-19 lockdown. Our results may not be generalizable to other age groups and further research should be carried out to assess what, if any, persistent effect the lockdown might have had on other school-aged students. If conditions allow, it would be ideal to have repeated measures over time from each participant to access changes in screening scores in further research. Assessing different outcomes based on differing levels of lockdown or self-isolation would also be a potential area for exploration. Self-reported data are subject to bias. Students completing a mental health screener may have underscored their responses to avoid being identified or to avoid further screenings. If students were afraid to be identified for depression, it is possible that our results underestimate the true effect the pandemic may have on adolescent depression.

Other studies that used data collected via social media or at a single point in time, make it difficult to evaluate changes in depression scores over time in similar samples of individuals. We are providing a unique insight into the magnitude of change in depression screening scores potentially due to the pandemic shutdown. Previous studies also completed their data collection a few months after the initial lockdown. Our data are also unique given the baseline screenings collected in the weeks leading up to lockdown protocols going into effect. Uncertainty around the pandemic was most likely more prevalent and coping strategies less so during those early months after lockdown restrictions were implemented.

There are several potential reasons why the mental health of high school athletes deteriorated, including reduction in physical activity, lack of community and familial support, or vulnerable socioeconomic positions [15,20,21,22]. Although we identified a change in the PHQ scores among high school student athletes, we are unable to identify specifically which factor might have been most harmful. Level of teammate camaraderie, access to physical training outside of school, community support, and/or home environment status are potential areas for future research.

Expansion to other high school students may be warranted to understand how the pandemic or other events are affecting high school students’ mental health. These results support the necessity of vigilant attention on the mental health and wellbeing of high school students, particularly given the interruption of high school athletics post COVID-19 lockdown.

## 5. Conclusions

The timing of these initial screenings provided a unique opportunity to examine depression risk immediately before and after pandemic countermeasures went into effect. Further understanding of adolescent athlete depression can facilitate the needed response to address this growing need. When high school athletes were separated from the benefits provided by organized school sports, such as physical activity level and social interactions, we saw an increase in the number of athletes scoring higher on depression screeners. Implementing mental health screening into routine physicals helped identify those potentially in need of early depression treatment in response to outside influences on mental health status.

## Figures and Tables

**Table 1 ijerph-19-09943-t001:** Summary of high school athlete depression screening participant demographics.

	Spring 20	Fall 20	Fall 21	Spring 22
Characteristics	*n* (%)	*n* (%)	*n* (%)	*n* (%)
PHQ-2 results by Gender				
Male	230 (59.7)	107 (73.8)	121 (74.2)	113 (48.3)
Female	155 (40.3)	38 (26.2)	40 (24.6)	111 (47.4)
Non-Binary	0	0	0	6 (2.6)
Prefer not to reply	0	0	0	3 (1.3)
Female and Non-Binary	0	0	2 (1.2)	1 (0.4)
PHQ-2 by Ethnicity				
Non-Hispanic	261 (67.8)	103 (71.0)	83 (50.9)	130 (55.6)
Hispanic	124 (32.2)	42 (29.0)	55 (33.7)	87 (37.2)
Prefer not to reply	0	0	6 (3.7)	2 (0.8)
Blank/Did not answer	0	0	19 (11.7)	15 (6.4)
PHQ-2 by Race				
African American/Black	58 (15.1)	14 (9.65)	25 (15.34)	35 (14.96)
Asian	18 (4.7)	2 (1.38)	6 (3.68)	21 (8.97)
Caucasian	140 (36.4)	55 (37.93)	69 (42.33)	98 (41.88)
Missing/Did Not specify	97 (25.2)	31 (21.38)	21 (12.88)	35 (14.96)
Multiple Race	56 (14.5)	30 (20.69)	31 (19.02)	35 (14.96)
Native American	1 (0.3)	0	1 (0.61)	2 (0.85)
Pacific Islander	15 (3.9)	13 (8.97)	10 (6.14)	8 (3.42)
PHQ-2 by Number of Sports				
Single	378 (98.2)	140 (96.6)	160 (98.2)	226 (96.6)
Multiple	7 (1.8)	5 (3.5)	3 (1.8)	8 (3.4)
PHQ-2 results by Grade				
9th	89 (23.1)	67 (46.2)	66 (40.5)	71 (30.3)
10th	111 (28.8)	32 (22.1)	54 (33.1)	70 (29.9)
11th	101 (26.2)	19 (13.1)	30 (18.4)	50 (21.4)
12th	81 (21.1)	27 (18.6)	11 (6.8)	43 (18.4)
Blank/Did not answer	3 (0.8)	0	2 (1.2)	0
PHQ-2 results by Age				
13	1 (0.3)	1 (0.7)	5 (3.1)	0
14	58 (15.1)	58 (0.40)	54 (33.1)	43 (18.4)
15	113 (29.3)	38 (26.2)	60 (36.8)	76 (32.5)
16	93 (24.2)	19 (13.1)	33 (20.2)	50 (21.4)
17	86 (22.3)	27 (18.6)	11 (6.8)	50 (21.4)
18	34 (8.8)	2 (1.4)	0	14 (5.9)
Blank/Did not answer	0	0	0	1 (0.4)
PHQ-9 results by Gender				
Male	10 (40.0)	12 (66.7)	6 (42.9)	5 (21.7)
Female	15 (60.0)	6 (33.3)	7 (50.0)	13 (56.5)
Non-Binary	0	0	0	2 (8.7)
Prefer not to reply	0	0	0	2 (8.7)
Female and Non-Binary	0	0	1 (7.1)	1 (4.4)
PHQ-9 by Ethnicity				
Non-Hispanic	17 (68.0)	10 (55.6)	4 (28.6)	9 (39.1)
Hispanic	8 (32.0)	8 (44.4)	6 (42.9)	13 (56.5)
Prefer not to reply	0	0	1 (7.1)	1 (4.4)
Blank/Did not answer	0	0	3 (21.4)	0
PHQ-9 by Race				
African American/Black	5 (20)	2 (11.1)	1 (7.1)	3 (13)
Asian	1 (4)	.	.	4 (17.4)
Caucasian	7 (28)	8 (44.4)	8 (57.1)	12 (52.2)
Missing/Did Not specify	7 (28)	4 (22.2)	2 (14.3)	0
Multiple Race	3 (12)	2 (11.1)	2 (14.3)	4 (17.4)
Native American	0	0	1 (7.1)	0
Pacific Islander	2 (8)	2 (11.1)	0	0
PHQ-9 by Number of Sports				
Single	25 (100)	18 (100)	14 (100)	19 (82.6)
Multiple	0	0	0	4 (17.4)
PHQ-9 results by Grade				
9th	5 (20.0)	6 (33.3)	5 (35.7)	10 (43.5)
10th	8 (32.0)	8 (44.4)	5 (35.7)	8 (34.8)
11th	6 (24.0)	1 (5.6)	3 (21.4)	2 (8.7)
12th	6 (24.0)	3 (16.7)	1 (7.1)	3 (13)
PHQ-9 results by Age				
13	1 (4.0)	0	0	0
14	2 (8.0)	6 (33.3)	2 (14.3)	6 (26.1)
15	8 (32.0)	7 (38.9)	7 (50.0)	8 (34.8)
16	5 (20.0)	2 (11.1)	4 (28.6)	5 (21.7)
17	6 (24.0)	2 (11.1)	1 (7.1)	3 (13.0)
18	3 (12.0)	1 (5.6)	0	1 (4.4)
Blank/Did not answer	0	0	0	0

**Table 2 ijerph-19-09943-t002:** PHQ screening results distribution by score category.

	Spring 20		Fall 20		Fall 21		Spring 22	
	*n* (%)	*Mdn* [25th, 75th]	*n* (%)	*Mdn* [25th, 75th]	*n* (%)	*Mdn* [25th, 75th]	*n* (%)	*Mdn* [25th, 75th]
PHQ-2								
0–2	363 (94.3)	0 [0, 1]	125 (86.2)	0 [0, 1]	149 (91.4)	0 [0, 1]	212 (90.6)	0 [0, 1]
3–6	22 (5.7)	3 [3, 5]	20 (13.8)	3 [3, 4]	14 (8.6)	3 [3, 4]	22 (9.4)	3 [3, 4]
PHQ-9								
0–4	6 (24.0)	2.5 [0, 3]	1 (5.6)	2 [2, 2]	0	0	2 (8.7)	4 [4, 4]
5–9	10 (40.0)	7 [6, 9]	9 (50.0)	7 [6, 9]	6 (42.9)	6 [5, 8]	7 (30.4)	7 [6, 8]
10–14	2 (8.0)	11.5 [11, 12]	3 (16.7)	10 [10, 14]	8 (57.1)	11.5 [10.5, 13]	6 (26.1)	11.5 [11, 13]
15–19	3 (12.0)	16 [15, 17]	5 (27.8)	17 [17, 19]	0	0	7 (30.4)	16 [15, 18]
20–27	4 (16.0)	22.5 [21, 23]	0	0	0	0	1 (4.4)	20 [20, 20]

**Table 3 ijerph-19-09943-t003:** Logistic regression analysis of predictors for scoring 3 or higher on PHQ-2.

Variables	OR	95% CI
Pre-COVID Comparison		
Spring 2020		
Fall 2020	**3.7**	[1.8, 7.6]
Fall 2021	1.6	[0.7, 3.5]
Spring 2022	1.2	[0.6, 2.4]
Sport Type		
Individual		
Team	0.7	[0.4, 1.2]
Ethnicity		
Non-Hispanic		
Hispanic	**1.5**	[1.0, 2.0]
Gender		
Male		
Female	**2.2**	[1.4, 3.7]

Values in bold are significant at *p* < 0.05.

## Data Availability

The data are not available due to privacy and ethical restrictions.

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
