# Peer review of "The Effect of COVID-19 Lockdown on PHQ Depression Screening Scores for High School Athletes"

_ijerph, 2022, doi:10.3390/ijerph19169943_

Round 1
Reviewer 1 Report
1.This study intends to argue a causal relation between depression screening scores for high school athletes and the COVID-19 Lockdown with an over-simplified research design, however, it's problematic.
2.First of all, it needs a panel data to record the level of depressive symptoms of these athletes at each observation period. In that case, to compare the change scores of depression screening of these athletes before and after the COVID-19 lockdown can make meaningful sense. However, for current situation of this paper (NT1=385, NT2=145, NT3=163, NT4=234), the sample size in each observation period are all vary from each other, it makes no sense that just simply comparing the change of level of depressive symptoms of high school athletes among these four period.
3.It is suggested that the author(s) only keep those high school athletes who had been participated in all of the four obversation period and re-do the analyses. In that case, the "real"change might be able to be watched.
4.Based on the research design of this study, there is no chance that this study is able to detect the argued causal effect of COVID-19 Lockdown upon the depressive symptoms of high school athletes since it did not control for the co-variated effects.
5.Apparently, this study needs to redo the research design, redo the data coding, and redo the data analysis before it could reach any decisive conclusion.
Author Response
- Regarding concerns about the variation in sample size, as stated in the manuscript (line 51), students were able to attend these screenings at their own discretion. To provide further clarification regarding this aspect of our study we have highlighted it again in the discussion section (line 189). The discussion also notes that reduced numbers post lockdown may have been due to concerns about large gatherings (lines 189-191). To avoid violating statistical test assumptions, we randomly selected 1 score from students who attended multiple sessions, which further decreased our sample size. This is clarified in our methods (line 101).
- Your suggestion of restricting analysis to students who participated in all 4 of the screening periods would have resulted in an n of 9 which we felt ignored important data.
- With regard to study design not being adequate to attribute changes primarily to the COVID-19 lockdown, we agree that this was a limitation of the study and had not implied a causal association in our manuscript. Additional changes have been made (line 155) to further reduce any misconceptions about strong claims of causality. The reasons school athletics were cancelled for the students in this study was due to the COVID 19 lockdowns. We believe that not referencing the timing of the screenings, and subsequent score changes would be an oversight in our discussion. We would have preferred a larger data set with additional variables but were limited to the data that was available.
Reviewer 2 Report
Dear authors of the article
I have reviewed your manuscript and find it an interesting and relatively understudied proposal within the field of COVID-19 about which much has been written in recent times. I believe that you contribute relevant information to the scientific literature on the subject. However, I also find several areas of opportunity in your research:
- In the introduction the authors point out that: "the effect these mandates had, and continue to have, on mental health are still being investigated and remain poorly understood." (rows 27-29), although afterwards they mention a series of studies that clearly refer to mental health effects in the form of anxiety, depression and stress. The scientific literature abounds with reports of mental health effects due to compulsory confinement and social distancing.
- The population group studied (high school athletes) is a relevant element of the work, since in fact there is not much research in this regard, however, I consider that works that can serve as a reference for the authors have been left out of the introduction/discussion, for example: https://doi.org/10.4085/478-20 ; https://doi.org/10.4085/1062-6050-0121.21 ; or https://doi.org/10.4085/1062-6050-0739.20
- In the material and methods section, the information could be better organized using subheadings to clearly differentiate the measurements, procedure, statistical analyses and ethical considerations.
- I consider especially important that the authors include the ethical considerations section within the manuscript and not just as a statement at the end of the paper. Please indicate if the work was approved by a local or international research ethics committee, if so, make explicit the identification code of the project. If not, please indicate whether the participants gave written or oral informed consent. Also include the adherence of your work to the Helsinki declaration as revised in 2013.
- the reliability of a test indicates the extent to which individual differences in test scores are attributable to random measurement error and the extent to which they are attributable to actual differences in the characteristic or variable being measured. The manuscript does not present information on the internal consistency of the instruments used in this population. Do the authors have information on the reliability coefficients of the PHQ-2 and PHQ-9 in the sample studied? Did they use any other type of reliability?
- The discussion section needs a lot of work; I recommend the authors to point out the differences or similarities of their results in comparison with those existing in the scientific literature to generate more contrast and relevance to their findings.
- In the conclusion, please include whether you consider that your results have practical implications for possible adoption by the target audience. Should there be policy recommendations from this research on mental health, education and/or physical activity?
Author Response
- We made a change in line 27-29 to address the comment regarding this line.
- In regard to suggested references:
- https://doi.org/10.4085/478-20 - was already included (reference #24)
- https://doi.org/10.4085/1062-6050-0121.21 - has been added (reference #26)
- https://doi.org/10.4085/1062-6050-0739.20 - similar to reference #24 – declined to include
- We have added subheadings to the materials and methods section to further organize the section
- Our study was deemed exempt by our IRB since it utilized previously collected data. Our IRB adheres to all laws and regulations required to monitor biomedical research involving human subjects. From our understanding this journal requires the ethical statements to be included at the end of the manuscript.
- We have added information about the PHQ-2 and PHQ-9 internal consistency in lines 64 & 71 respectively in response to your comment about the reliability of the test.
- Regarding a comparison to other studies within our discussion, we have already included that in line 180 and believe it to be sufficient.
- We have expanded the conclusion to include a few statements regarding the real-world implications that may be derived from our study in response to your comment (lines 221-226).
Reviewer 3 Report
Reviewer Report
This study examined depression among high school athletes before and during the COVID-19 pandemic using a self-administered questionnaire. The study reported that the COVID-19 pandemic impacted high school athletes. It is interesting to note that post-pandemic measurements have also been examined for mental resilience. The results of this study may serve as reference material when implementing mental guidance for high school athletes.
Major Comments
An essential point in this study is a few measurements were conducted from the spring of 2020 to the spring of 2022, and changes in the results are discussed. The timing of each measurement used within this study is somewhat broad terms, such as before, during, and after the pandemic. It is essential to clarify the pandemic status of COVID-19 at the time of each survey and the situation at the time of the study. Therefore, the pandemic status should be presented by the period in which each survey was conducted, for the region in which the survey was conducted.
Please clarify the definition of the target population, high school athletes. Please reexamine whether simply being a member of an athletic team in high school is a requirement.
In this study, the subjects were limited to high school athletes, and the measurement results were examined. I believe that the purpose of this study can be better established by clarifying comparisons between high school athletes and non-high school athletes in previous studies and other studies involving high school students.
Please clarify the status of athlete activities in high schools during the pandemic. I believe it will be essential to provide numerical data on whether activities were suspended and for how long they were suspended.
Author Response
Thank you for your comments and suggestions, please see our response below:
- For pandemic status at each screening period, we added a subheading (2.2.1) within the methods and provided more context around the status of COVID at each screening timepoint.
- We added a clarification of the population in the methods section (line 75)
- Regarding the limitation of subjects to high school athletes, since the data set we used came from a school sponsored event for only those participating in school sports we had no data on non-high school athletes. We do highlight in our introduction (lines 44-46) the differences that physical activity and sports participation have on youth mental health. While a full comparison to non-high school athletes is a good idea, we believe this is an idea better intended for a separate manuscript comparing athletes vs non-athletes
Round 2
Reviewer 1 Report
1.This revised version made a tremendous improvement especially in arguments of its significant findings. Avoiding causal arguments are certainly a good move for this revised paper.
2.It is suggested that this paper provide some more information about an ideal research design for this kind of study that can provide more insightful research findings for readers.
Author Response
- Thank you for your comments regarding the improvements in revisions submitted.
- We added a further clarification in lines 184-187 regarding ideal study design for future research.
Reviewer 2 Report
Dear authors of the article
1- You have added information on the internal consistency reported by other studies; however, you still do not present an internal consistency coefficient (Cronbach's alpha in this case since PHQ-2 and PHQ-9 are Likert scales) for your sample. It is important that you add this information to the final version of your manuscript.
2- The discussion has no relevant changes since there is practically no contrast between other studies and your results. Relevant studies and its implications considering your results are not cited. I recommend you cite the papers with which your findings can be contrasted.
3- The authors point out that: “Our findings are limited to a single high school and, therefore, may not be generalizable to other schools. Some students may have received their school athletic physical from their own physician, so these results may not be representative of all student athletes at this or other schools.” (Rows 177-179) These seems to be a very serious limitation. Then you stated: “Expansion to all high school students may be warranted to understand how the pandemic or other events are affecting high school students’ mental health. These results support the necessity of vigilant attention on the mental health and wellbeing of high school students particularly given the interruption of high school athletics post COVID-19 lock-down.” (Rows 203-207)
Authors should clarify whether their results have implications outside the high school where the research was conducted.
4- Authors also said: “Self-reported data are subject to bias. Students completing a mental health screener may have underscored their responses to avoid being identified or to avoid further screenings.” (Rows 184-185). I recommend that you also provide suggestions for future studies to address these limitations.
5- “There are several potential reasons why the mental health of high school athletes deteriorated, including reduction in physical activity, lack of community and familial support, or vulnerable socioeconomic positions.” (Rows 196-198) What makes you think that? For this reason, it is important to cite other works.
Author Response
- We are not quite sure what variables the reviewer would believe we should run a Cronbach’s alpha within our data. Since the PHQ-2 is comprised of the first two questions of the PHQ-9, this would violate statistical assumptions required for the Cronbach’s alpha test. The PHQ depression screens are validated tools and therefore should not require additional independent validation within this study. Our approach of citing the validation studies is also present in the same literature this reviewer suggested in their first round of comments. If there is confusion regarding the fact that the PHQ-2 is comprised of the first two questions of the PHQ-9, a clarification has been added to line 60
- Lines 194-202 discuss the differences between our dataset and data used in previous literature. We believe these points to be the biggest difference to other studies. A sentence further highlighting this difference has been added in line 198. We explicitly recognize the similarity of our results regarding mental health decline to previous literature in lines 173-177.
- While it is a limitation, it is the data that were available. Having baseline data weeks before the onset of the pandemic affords us a unique opportunity to have insight into what might have been going on in high school athletes’ minds. We feel that this benefit outweighs the limitation of single site data and should be shared. We have further highlighted this point with the addition in line 198.
- Suggestions for further studies has been increased in lines 185-188.
- This is in reference to the introduction, lines 33 & 43, where these factors have all been identified from previous studies as having influences on adolescent mental health. Citations have been added to further clarify.